# 14-year-old Schoolchildren can Consent to Get Vaccinated in Tyrol, Austria: What do They know about Diseases and Vaccinations?

**DOI:** 10.3390/vaccines8040610

**Published:** 2020-10-15

**Authors:** Peter Kreidl, Maria-Magdalena Breitwieser, Reinhard Würzner, Wegene Borena

**Affiliations:** 1Department for Hygiene, Microbiology and Public Health, Medical University Innsbruck, 6020 Innsbruck, Austria; peter.kreidl@i-med.ac.at (P.K.); reinhard.wuerzner@i-med.ac.at (R.W.); 2Department of Internal Emergency Medicine, Klinikum Wels-Grieskirchen, 4600 Wels, Austria; MariaMagdalena.Breitwieser@klinikum-wegr.at; 3Institute of Virology, Department for Hygiene, Microbiology and Public Health, Medical University of Innsbruck, 6020 Innsbruck, Austria

**Keywords:** vaccination, vaccine hesitancy, adolescent, measles, mumps rubella, human papilloma virus, informed consent

## Abstract

In Austria, consent to receiving vaccines is regulated at the federal state level and in Tyrol, children aged 14 years are allowed to consent to receiving vaccination. In August 2017, we investigated determinants associated with vaccine hesitancy, having been vaccinated against measles and human papillomavirus (HPV) and the intention to vaccinate among schoolchildren born in 2002 and 2003. Those who consider measles and HPV a severe disease had a significantly higher intention to be vaccinated (prevalence ratio (PR) of 3.5 (95% CI 1.97–6.32) for measles and a PR of 3.2 (95% CI 1.62–6.35) for HPV). One-third of the participants (32.4%; 95% CI 27.8–37.4) were not aware that they are allowed to consent to receiving vaccines. The most common trusted source reported by respondents (*n* = 311) was the medical doctor (80.7%; 95% CI 75.7–84.7). The main finding related to the aim of the study was that the proportion of objectors is below 4% and therefore it should still be possible to reach measles elimination for which a 95% uptake is necessary. Although the proportion of objectors is not higher compared to adults, we recommend to intensify health education to increase health literacy.

## 1. Introduction

In 2019, vaccine hesitancy (VH) was stated as one of the ten key health threats by the World Health Organization (WHO) [1]. The majority of studies investigated VH among parents and health care providers [2,3,4], but little is known on the knowledge, perceptions, attitudes and practices of adolescents and young adults who will be future parents. In recent years, in several Western European countries, the highest incidence of several vaccine preventable diseases (VPD) such as measles, mumps or pertussis was reported among adolescents [5]. In Austria, all childhood vaccinations are recommended including influenza and most of them free of charge until age (except vaccination against meningococcal B, tick-borne encephalitis, varicella and influenza vaccination) [6].

Health literacy is an important determinant of empowerment and decision-making including decisions about vaccinations. In a survey that assessed the level of health literacy in a population among several European countries, Austria had one of the lowest literacy indexes [7]. In Austria, educational curricula for schoolchildren neither include information on vaccine preventable diseases (VPD) nor on the safety and effectiveness of vaccination and its impact [8]. Youth protection laws are the responsibility of each federal state in Austria. Adolescents aged 14 years and above (attending grade 8 and grade 9 school classes) are legally allowed to consent on vaccination on their own in Tyrol [9]. This combination of a poor health literacy index and the freedom to decide on one’s own immunization status at a very young age motivated us to investigate several aspects of immunization behavior among this age group.

The main aim of this cross-sectional study was to describe the knowledge, perceptions, vaccination status and intention to vaccinate among the youngest of these adolescents. We aimed to assess the level of VH and factors responsible for that at this early stage of self-determined life of young individuals. The information obtained from this survey will be a useful tool in tailoring immunization strategies in the region and could also be useful for other regions facing a similar situation. 

## 2. Materials and Methods

### 2.1. Study Population

The target populations of this study were children/adolescents aged 14 years attending eighth and ninth grade school classes in Tyrol. In Tyrol, one of Austria’s nine states with 745,049 inhabitants [10], the number of students attending those classes in 210 schools was 13,819 in 2017 [9]. 

### 2.2. Study Design and Sampling Strategy

A sample size of 374 participants was calculated to assess a prevalence with +/− 5% precision [11]. Taking a population size of 13,819 14-year-olds, we used a sample-proportional-to-size design [12] to select participants and assumed a worst-case scenario of a 50% prevalence of knowledge factors. Only children born in 2002 or 2003 were included in the study.

The school authorities provided aggregated lists with all 210 classes with schoolchildren in grades 8 and 9 attended by children born in 2002 or 2003. This list of eligible classes including the number of schoolchildren was sorted alphabetically. The cumulative number of schoolchildren and subsequently a sampling interval for 30 clusters were calculated (sampling interval = 461). A random number between one and the sampling interval was chosen (random number = 132) and thirty classes were identified. Each entire class was recruited in which a systematically selected schoolchild was part of. All schoolchildren of a selected class were included resulting in 2443 potential participants. For each school, two substitute school classes were selected in case of non-response. 

The headmasters of all 30 selected schools were contacted by telephone and email and were encouraged to support the participation of schoolchildren in the study. A facilitation letter from the regional school authority to enhance participation was attached. Additionally, all school physicians of selected schools were briefed and supported our activities.

### 2.3. Questionnaire and Data Collection

The potential participants were informed by the responsible teachers of each class about the study and received an information letter including the informed consent which they had to bring home to their parents, who were encouraged to sign it. Schoolchildren then brought the signed informed consent back to school. After having obtained informed consent from the parents, data were collected between June 2017 and May 2018 using an anonymized, web-based standardized self-administered questionnaire. Data were entered in a password-protected survey tool, which could be accessed from personal mobile phones or by using computers in dedicated rooms in the schools [6]. Data collection took place during school hours to increase the participation rate. 

The survey contained 42 questions about knowledge, attitudes and intention to vaccinate and history of measles mumps rubella (MMR) and human papilloma virus (HPV) vaccine uptake. Personal information contained sex, birth cohort, school type, migratory background and residential district.

The standardized questionnaire was adapted from available survey tools [13,14] and pilot-tested with pupils of the same age to ensure high comprehensibility. Several questions were coded in 5 point Likert scales such as “not being against vaccinations in general but deciding for each vaccination separately”, “vaccinations are mainly the interest of pharmaceutical companies” or “I am not interested in the discussion about vaccinations”. For these questions, the inconclusive category (neither agree nor disagree) was excluded to estimate the association with self-reported vaccination and the intention to vaccinate. Partially missing data were excluded from the analysis.

The study was conducted in accordance with the Declaration of Helsinki and the protocol was approved by the Ethics Committee of the Medical University Innsbruck (EK 1072/2017) and the regional school authority.

### 2.4. Data Entry and Analysis

The anonymous data were extracted from the web-based data entry tool and analyzed in Epi.info^TM^ version 7.2.2.2 (CDC, Atlanta).

We described the distribution/prevalence of the answers for each question excluding missing answers. We compared the prevalence of knowledge between questions using a chi square test and considered a value below 0.05 as statistically significant. We estimated the prevalence ratio and the 95% confidence interval of potential determinants using the Taylor series in Epi.info^TM^ between the vaccinated and unvaccinated group using univariate analysis. Exposures were determinants such as sex, migratory background, preference of alternative medicine or VH and outcome was defined as reported MMR or HPV vaccination or intention to vaccinate against MMR or HPV.

### 2.5. Definitions

Vaccinated: recall of at least one dose of MMR or HPV;Unvaccinated: reported not having received at least one dose of MMR or HPV;Unknown vaccination status: non-responders, persons who did not recall or did not want to answer the question about previous MMR or HPV vaccinations;Migratory background: either born outside Austria or at least one parent born outside Austria.

## 3. Results

### 3.1. Characteristics of Study Participants

The questionnaires were completed in an anonymous way and only the school type was recorded, therefore no response rate by class or school could be calculated. 

Of the 518 respondents, 367 participants were born in 2002 or 2003 and thus eligible for inclusion in our study (Figure 1). 

Of the 367 participants, more than half were born in 2002. Fifty-eight percent of participants were males. Participants were residents from eight of the nine Tyrolean districts. The only district without participants was the smallest Tyrolean district accounting for 4% of the birth cohorts of interest. On average, 2.2% of eligible schoolchildren participated in our survey, ranging from 0.2% to 4.5% by district. Participants from the four districts with the highest number of eligible schoolchildren accounted for 91.7%. These districts were also the ones with the highest number of schools per inhabitant. The majority of participants were born in Austria, and 1/3 reported a migratory background (Table 1).

### 3.2. General Knowledge on Vaccinations

The majority of participants knew about the success of vaccinations to reduce the burden and mortality of communicable diseases (82.6%; 95% CI 78.2–86.2). Two-thirds disagreed with the statement that the risks of vaccination outweigh the benefits (65.4%; 95% CI 60.2–70.3). The mechanism of live vaccines was known by more participants (45.8%; 95% CI 40.6–51.1) compared to the mechanism of inactivated vaccines (21.7%; 95% CI 17.7–26.3) (*p* < 0.0001). The fact that smallpox has been eradicated was known by one-third of participants (31.1%; 95% CI 26.4–36.2). Only one-third (34.9%; 95% CI 30.0–40.1) of participants were aware that they did not need their parents’ consent to get vaccinated when aged 14 years and older.

### 3.3. Knowledge about Measles Disease and Vaccination

Most participants (79.4%; 95% CI 74.7–83.3) knew about the high transmissibility of measles and that measles can infect anybody who is not immune irrespective of age (68.0%; 95% CI 62.8–72.4). Nearly half of the participants (45.1%; 95% CI 39.9–50.5) agreed with the statement that measles can be acquired several times during life and does not provide life-long immunity. More than half of the participants were aware that measles cases and outbreaks frequently occur in Austria (56.6%; 95% CI 51.2–61.7) and 35.5% (95% CI 30.7–40.6) knew that MCV is a live attenuated vaccine. Less than a third of participants (28.8%; 95% CI 24.3–33.8) knew that the vaccine effectiveness of measles vaccination is above 80%.

### 3.4. Knowledge about HPV Disease and Vaccination

More than half of the participants (55.1%; 95% CI 49.7–60.3) disagreed that HPV is a childhood disease and knew that it is mainly transmitted via sexual contact (62.4%; 95% CI 57.1–67.4). Cervical cancer caused by HPV was known by twice as many participants (40.8%; 95% CI 35.8–46.0) compared to oropharyngeal cancer (18.4%; 95% CI 14.7–22.8) (*p* < 0.0001) and more than a quarter knew that HPV infection can either cause genital warts (28.0%; 95% CI 23.6–33.0) or present as a subclinical infection (26.9%; 95% CI 22.6–31.8). Around two-thirds of the participants knew (67.2%; 95% CI 62.0–72.0) that HPV vaccination is recommended for both sexes, and provided in schools, free of charge, for children between 9 and 12 years of age (27.6%; 95% CI 23.1–32.6). One in five participants knew (22.1%; 95% CI 18.0–26.8) that the vaccine effectiveness of HPV vaccine is above 80%.

### 3.5. Perception of the Severity of Diseases

Poliomyelitis was perceived most frequently as severe or rather severe VPD followed by HPV, tetanus, tick-borne encephalitis (TBE), diphtheria and mumps. Measles ranked at position seven prior to rubella and pertussis. Influenza was considered by less than a third of the participants as a severe or rather severe disease (Table 2).

### 3.6. Intention to Vaccinate

The intention to vaccinate was highest for TBE and lowest for influenza and similar, but not identical to the perception of the severity of diseases. The highest association between the perception of severity and intention to vaccinate was for TBE followed by measles and HPV (Table 3).

### 3.7. Vaccine Hesitancy

Less than four percent (3.6%, *n* = 12; 95% CI 2.1–6.2) of respondents fully agreed with the statement that “vaccinations are unnecessary and harmful” and were therefore classified as objectors, whilst 19.8% (95% CI 15.9–24.4) were classified as hesitant (partly agreed, or neither agreed nor disagreed) and 76.6% (95% CI 71.7-80.8) as pro-vaccination.

Nearly half of the participants (42.6%; 95% CI 37.4–48.1) completely or partly disagreed with the statement that “vaccinations are mainly of interests of pharmaceutical companies”, 21.0% (95% CI 17.0–25.7) agreed and 36.3% (95% CI 31.4–41.6) neither agreed nor disagreed.

The majority of participants completely or partly agreed (74.5%; 95% CI 69.5–78.7) with the statement that they are not against vaccinations in general but would make a “selective decision for each vaccination”, 16.5% (95% CI 12.9–20.9) completely or partly disagreed and 9.0% (95% CI 6.4–12.6) were undecided.

More than two-thirds (69.7%; 95% CI 64.5–74.4) of the participants completely or partly agreed in having “interest in the discussion about vaccination”, 15.9% (95% CI 12.4–20.2) completely or partly disagreed and 14.4% (95% CI 11.1–18.6) were undecided (Figure 2).

Among the twelve respondents classified as objectors, three of seven who provided information reported having been vaccinated against measles, but none against HPV.

### 3.8. Reported MMR and HPV Vaccination Status 

More than half of the 367 participants, 51.8 (95% CI 46.7–56.8), did not want to respond or did not recall their vaccination status related to measles and even more related to HPV (65.4%; 95% CI 60.4–70.1). Among respondents, the estimated proportion having been vaccinated against measles was 83.1% (95% CI 76.7–88.3) and 38.6% (95% CI 30.1–47.6) against HPV. The vast majority of participants did not recall the number of doses received against measles (83.7%; 95% CI 79.5–87.1) or HPV (91.0%; 95% CI 87.6–93.5). More girls compared to boys recalled the number of received doses of either vaccine (PR 1.28; 95% CI 1.13–1.44).

The proportion of females who recalled having been vaccinated against measles was slightly higher compared to males (87.6%) versus (79.3%) (PR = 1.3: 95% CI 0.95–1.84) and so was the recalled HPV uptake (31.6% of males and 44.1% of females vaccinated) (PR 1.4; 95% CI 0.89–2.07).

### 3.9. Determinants of Vaccine Hesitancy

#### 3.9.1. Demographic Factors

The PR for females being pro-vaccination was 1.2 (95% CI 0.95–1.44) and for not having a migratory background, it was 1.3 (95% CI 0.91–1.88), therefore neither sex nor migratory background were associated with hesitancy.

#### 3.9.2. Perception of the Severity of Diseases

The association (PR) between the perception of the severity of disease and having been classified as pro-vaccination was significant for polio, HPV, tetanus and diphtheria (Table 4).

The PR between the perception of the severity of measles and recalled measles vaccination was 1.8 (0.93–3.42) and thus higher compared to the association of the intention to vaccinate. This was the opposite for HPV (PR = 1.7; 95% CI 0.83–1.65).

#### 3.9.3. Having Been Vaccinated against Measles and HPV and the Intention to Vaccinate 

Classified as hesitant (objectors excluded) was associated, with a PR of 3.7 (95% CI 1.91–7.23), with not being vaccinated against measles, and a PR of 1.6 (95% CI 1.18–2.01) with not having been vaccinated against HPV, compared to the pro-vaccination group.

The PR of hesitant persons not intending to vaccinate against tetanus was the highest, followed by TBE and HPV. Measles was ranked at position seven (Table 5). 

#### 3.9.4. Other Factors

Agreement with the statement to take a selective decision for each vaccination was similar between the hesitant and the pro-vaccination group (PR = 1.1; 95% CI 0.63.3.52). No interest in discussions about vaccination was strongly associated with hesitancy (PR = 5.5; 95% CI 3.41–9.00) as well as agreeing with the statement that vaccinations are mainly the interest of pharmaceutical companies (PR = 2.9; 95% CI 2.06–4.15).

### 3.10. Other Determinants of Having Been Vaccinated against Measles and HPV

More than one-third of participants (41.3% 95% CI 34.8–48.1) stated that they trust alternative medicine such as homeopathy, acupuncture or Ayurveda more compared to traditional medicine. More trust in alternative medicine was associated with not having been vaccinated against measles (PR 1.6, 95% CI 1.07 to 2.51) and HPV (PR 2.0, 95% CI 1.05 to 3.71). The results were similar when stratifying for migratory background and sex. 

The majority of participants reported to possess a vaccination card (97.4%; 95% CI 95.0–98.7). The reported possession of a vaccination card was strongly associated with having been vaccinated against measles (PR 22.5, 95% CI 2.61–193.07) and not calculable for HPV, as all HPV-vaccinated participants reported to possess a vaccination card.

### 3.11. Determinants of the Intention to Vaccinate

Preference of alternative medicine was associated with not intending to vaccinate against measles and mumps (Table 6). No other determinants such as sex or migratory background were significantly associated with the intention to vaccinate against any of the investigated diseases.

### 3.12. Information Sources Related to Vaccinations

Sixty-three (19.3%; 95% CI 15.4–23.9) of 327 participants who responded reported that they actively searched for information related to vaccination. Medical doctors and the family were by far the most common sources of information (Table 7).

## 4. Discussion

The unique aspect of this study was that we targeted young adolescents, namely 14 year olds. The significance lies in that this young population has the right to decide on their vaccination status in Austria. The main finding related to the aim of the study was that the proportion of objectors is below 4% and therefore it should still be possible to reach measles elimination for which a 95% vaccine uptake is necessary. The proportion of hesitant persons was less than 20% and strongly associated with not having been vaccinated against measles and HPV as well as the intention to vaccinate compared to the pro-vaccination group. 

More than half of the participants did not recall their self-reported MMR vaccination status. The reported MMR coverage was 83% and similar to the reported coverage [15]. The HPV reported coverage was 40% and much lower compared to measles despite the fact that our study population was a target group for the free of charge school program. Girls recalled their vaccination status more frequently than boys.

Considering a disease as severe was associated with a positive intention to vaccinate as well as possessing a vaccination card; preference of alternative medicine was negatively associated. These determinants were described previously as being associated with non-vaccination among adults [14,16,17,18,19,20]. Therefore, it is important to integrate disease-specific information in school curricula. 

Measles was perceived as the seventh most severe vaccine-preventable disease in our survey in contrast to TBE, which was considered as the second most severe disease by our participants. This lower perception of severity may have been partly due to the very active TBE vaccine promotion in all media and on billboards with ticks attacking humans in Austria. The estimated TBE vaccine coverage in Austria in 2018 was 85% [21] and thus higher compared to the coverage of measles. Without any doubt, TBE is a severe disease and since 2016, an increase in case numbers with 154 hospital-admitted patients in 2018 was reported in Austria. Fifty-two percent of them were classified as severe including five deaths [22]. Due to the high TBE vaccine uptake, the incidence of TBE declined in Austria from 4.9 cases per 100,000 inhabitants prior to the implementation of vaccination in 1981 to 1.9 per 100,000 inhabitants in 2018 [22,23].

The distribution of vaccine hesitancy (VH) was similar compared to a recently published national study of VH among parents of children aged 16 to 36 months in Italy [3]. Nevertheless, compared to the Italian study, we observed a higher proportion of objectors and hesitant participants but much lower compared to a study conducted in 2014 among 350 Austrian patients in an emergency department [24]. 

It was surprising that only one-third of participants knew about their right to decide to get vaccinated at age 14 [9] and that these minors could chose to be vaccinated over parental objections. This lack of awareness that schoolchildren do not need their parents’ consent to vaccinate was neither associated with vaccine hesitancy nor the intention to vaccinate against measles or HPV, but this may change if more schoolchildren are aware about their rights in the future.

In Austria, the second lowest level of health literacy among eight European countries was described in 2015, where more than half (56.4%) of the population was classified as having inadequate or problematic health literacy [7]. Without comparison to other countries or regions, it remains difficult to compare this specific aspect of health literacy. The right to decide at this young and the low health literacy are a detrimental combination which urgently needs attention from health policy makers. 

Calculation was defined by Betsch et al. as “individuals’ engagement in extensive information searching” [25] and can lead to non-vaccination. In our study, interest in discussion about vaccinations was positively associated with both the intention to vaccinate and reported vaccination against measles and HPV.

Trust in health care providers is a key determinant for vaccine uptake [26]. Only few participants reported having actively searched for information related to vaccination. Medical doctors were by far the most important and most trustworthy source, which is similar to published results for adults [27]. Smith et al. suggested that the effect of information sources on vaccine uptake needs further studies [28]. The responsibility of health professionals is vital with respect to actively recommending vaccination. 

One important limitation of our study was that the decision of schools to participate was mainly dependent on the headmaster’s agreement which may have resulted in selection bias. However, the very active role of school physicians in convincing headmasters and recruiting participants positively influenced the willingness of headmasters to participate in our survey.

Another important limitation of our study was that many participants did not respond to all questions. This limited the power of the study and thus could have masked other significant associations. Furthermore, we could not conduct a sensitivity analysis to investigate potential correlation structures within each class, as data on classes were not obtained. Generalizability is another limitation of our study as the study area represented only one region of Austria, however, the sample size was still sufficient to draw plausible conclusions.

We defined the vaccination status by history only and did not check the vaccination records due to logistical reasons. This may also have biased our results.

Following measles outbreaks in the United States in 2019, some affected children pursued vaccination against the parents’ resistance to vaccination such as Ethan Lindenberger [29]. He was only able to get vaccinated at the age of 18 years. In Tyrol, Austria, children could already decide at age 14 to get vaccinated without parental permission [9]. The insufficient coverage of many vaccines and repeated outbreaks triggered our survey to assess vaccine hesitancy among adolescents. The Society for Adolescent Health and Medicine suggests in a position paper that minor consent should be granted. Only one-third of participants were aware that they are allowed to consent to vaccination without parental permission. This fact and the lack of knowledge about the safety and effectiveness of nearly every fifth 14-year-old participant demonstrate that awareness raising in this age group could be an important instrument to increase coverage, as well as the low level of health literacy. 

Already two years prior to our study, we implemented school lectures targeting VPD and their respective vaccinations as we consider it very important that every child should be confronted at least once during her/his “school-life” with relevant communicable diseases and the safety and effectiveness of vaccinations, which are considered the most effective preventive measures besides safe drinking water and hygiene. We continue these activities now for several years and provided baseline lectures in easy to understand language to the Ministry of Education which can be used by any school doctor or biology teacher. Furthermore, transparent, easy to understand and pro-vaccination messages should be distributed by social media.

Furthermore, we use the European immunization week each year to raise awareness among the general population, where we work closely together with the Austrian Medical Student Association and other important stakeholders.

## 5. Conclusions

We conclude that the level of knowledge about vaccine-preventable diseases is limited in our region. We recommend a different strategy to approach adolescents and increase their knowledge about vaccine-preventable diseases.

## Figures and Tables

**Figure 1 vaccines-08-00610-f001:**
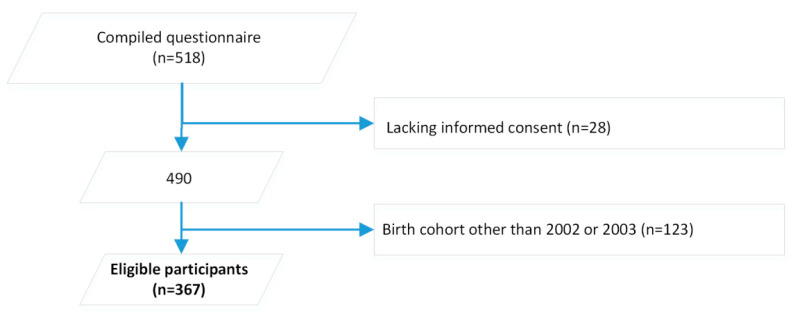
Number of participants after application of exclusion criteria (*n* = 367).

**Figure 2 vaccines-08-00610-f002:**
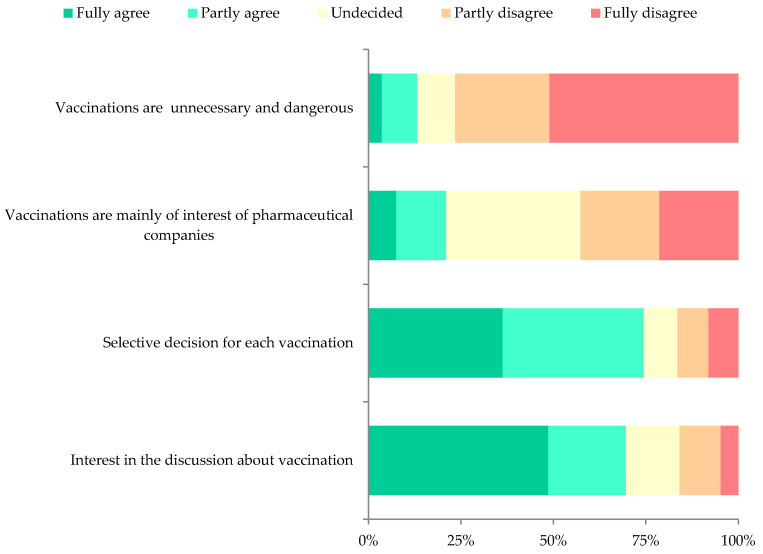
Distribution of vaccine hesitancy determinants.

**Table 1 vaccines-08-00610-t001:** Demographic characteristics of survey participants (*n* = 367).

Category	Sub-category	Number of Participants	Proportion of Participants (%)	95% Confidence Interval
Sex				
	Males	212	57.8	52.7–62.7
	Females	150	40.1	36.0–46.0
	unknown	5	1.4	0.6–3.2
Birth cohort				
	2002	210	57.2	52.1–62.2
	2003	157	42.8	37.8–47.9
Total population 2002/2003 by district	13,819		% of total	
3167	Innsbruck Land	143	39.0	34.1–44.0
2093	Innsbruck Stadt	66	18.0	14.4–22.2
2060	Kufstein	45	12.3	9.3–16.0
1609	Schwaz	79	21.5	17.6–26.0
1182	Imst	18	4.9	3.1–7.6
1181	Kitzbühel	3	0.8	0.3–2.4
1004	Lienz	2	0.5	0.2–2.0
958	Landeck	7	1.9	0.9–3.9
565	Reutte*	0	-	-
	unknown	4	1.1	0.4–2.8
Place of birth				
	Austria	339	92.4	89.2–94.7
	Outside Austria	20	5.5	3.6–8.3
	unknown	8	2.2	1.1–4.2
Migratory background				
	Both parents born in Austria	241	65.7	60.7–70.3
	One parent born outside Austria	57	15.5	12.2–19.6
	Both parents born outside Austria	56	15.3	11.9–19.3
	unknown	13	3.5	2.1–6.0

**Table 2 vaccines-08-00610-t002:** Perception of the severity of different vaccine-preventable diseases by study participants.

Disease	Severe or Rather Severe	95% CI
Polio	94.9%	91.8–97.1
HPV	87.8%	83.2–91.5
Tetanus	87.8%	83.2–91.6
TBE	86.9%	82.6–90.5
Diphtheria	71.4%	64.5–77.3
Mumps	68.2%	62.1–73.9
Measles	67.6%	61.9–72.9
Rubella	55.2%	49.1–61.3
Pertussis	52.7%	46.7–58.7
Influenza	32.2%	27.2–37.7

**Table 3 vaccines-08-00610-t003:** Association (prevalence ratio (PR) and 95% confidence interval (95% CI)) between the perception of the severity of disease and intention to vaccinate against the disease.

Disease	PR	95% CI
TBE	7.7	3.81–15.57
Measles	3.5	1.97–6.32
HPV	3.2	1.62–6.35
Diphtheria	3.1	1.62–6.00
Mumps	3.0	1.74–5.18
Pertussis	2.5	1.67–3.73
Rubella	2.3	1.42–3.84
Tetanus	2.1	0.85–5.02
Poliomyelitis	1.8	0.50–6.31
Influenza	1.4	1.08–1.83

**Table 4 vaccines-08-00610-t004:** Association (prevalence ration (PR) and 95% confidence interval (95% CI)) between the perception of the severity of disease and having been classified as pro-vaccine.

Disease	PR	95% CI
Poliomyelitis	2.3	1.14–4.77
HPV	2.3	1.34–3.86
Tetanus	2.3	1.34–3.86
Diphtheria	2.2	1.29–3.76
TBE	1.4	0.77–2.64
Influenza	1.4	0.80–2.54
Measles	1.3	0.82–2.16
Mumps	1.1	0.64–1.98
Rubella	1.1	0.67–1.76
Pertussis	0.9	0.54–1.50

**Table 5 vaccines-08-00610-t005:** Association (prevalence ratio (PR) and 95% confidence interval (95% CI)) between having been classified as hesitant and intention to vaccinate.

Disease	PR	95% CI
Tetanus	5.6	2.67–11.55
TBE	5.2	2.23–12.03
HPV	5.1	2.67–9.87
Diphtheria	3.9	2.16–7.07
Mumps	3.6	2.17–6.08
Measles	3.4	1.86–6.14
Pertussis	2.4	1.72–3.30
Rubella	2.3	1.44–3.74
Poliomyelitis	2.1	1.02–4.39
Influenza	1.4	1.15–1.70

**Table 6 vaccines-08-00610-t006:** Association (prevalence ratio (RR) and 95% confidence interval (95% CI)) of the preference of alternative medicine and intention to vaccinate.

Disease	PR	95% CI
Mumps	2.4	1.29–4.54
TBE	2.4	0.88–6.39
Measles	2.3	1.13–4.77
HPV	1.9	0.84–4.23
Pertussis	1.4	0.88–2.21
Rubella	1.4	0.82–2.42
Poliomyelitis	1.4	0.72–2.89
Tetanus	1.3	0.52–3.19
Diphtheria	1.2	0.52–2.59
Influenza	1.1	0.82–1.35

**Table 7 vaccines-08-00610-t007:** Most common trusted sources of information (*n* = 311).

Source of Information	Proportion	95% CI
Medical doctor	80.7%	76.0–84.7
Family	7.1%	4.7–10.5
Public health authority including ministry of health	5.5%	3.4–8.6
Pharmacy	3.5%	2.0–6.2
Internet and social media	1.9%	0.9–4.1
Friends	1.0%	0.3–2.8
General media (newspaper and TV)	0.3%	0.1–1.8

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
