# Peer review of "14-year-old Schoolchildren can Consent to Get Vaccinated in Tyrol, Austria: What do They know about Diseases and Vaccinations?"

_vaccines, 2020, doi:10.3390/vaccines8040610_

Round 1

Reviewer 1 Report

Manuscript Number: Vaccines-957598

Title: 14 year-old schoolchildren can consent to get vaccinated in Tyrol, Austria: What do they know about diseases and vaccinations?

Article Type: Article

Comments to the authors:

Summary: The paper evaluates the proportion of school children that decide for or against a particular vaccination and the importance of health education at schools to increase the health literacy and thereby acceptance to vaccination. This article is an important collection of information and of interest to the general public. I highly recommend the publication of this article in Vaccines as it is after a minor spell check.

L.71 correct the sentence please

Author Response

Response to reviewer 1:

The authors want to thank reviewer 1 very much the rapid and positive review process and recommendation for publication.

L71: The sentence has been corrected and now reads (Line 74 of the new manuscript): “Each class, in which the systematically selected school child was part of, was recruited for the survey.”

Reviewer 2 Report

In this manuscript titled “14 year-old schoolchildren can consent to get vaccinated in Tyrol, Austria: What do they know about disease and vaccination?”, Kreidl et al investigated the knowledge, vaccine perceptions, vaccination status and intent to receive vaccination among 14 year-old adolescents. The majority of findings were not particularly surprising although informative. Overall, the manuscript is easy to read and the presented data would be helpful in designing strategies to approach adolescents to increase their vaccination coverage. One interesting aspect of this study is the finding that the only one thirds of respondents knew that they had the right to consent to vaccines. Could the authors clarify if these minors can choose to be vaccinated over parental objections?  And if so, what would be the implication? Could the authors discuss this point? It will be interesting and important to investigate if granting 14-year-old school children the authority to receive vaccinations without parental permission would increase vaccination rate among families who oppose vaccinations. Since the proportion of objectors among respondents is comparable to adults, one would think that the vaccination coverage may not change significantly even if adolescents are aware of their authority to consent to vaccinations. However, this should be formally investigated. Another interesting follow-up study would be to investigate whether these respondents would change their responses to the questionnaires later i.e. one or two years after the completion of this survey. It is likely that learning that they have the authority to consent will increase their interests to learn more about vaccines and this may in turn change their attitude or perception towards vaccination. If data is available, could authors analyze the data to see if an association exist between awareness that they do not need their parents’ consent to get vaccination and their intent to vaccination and/or vaccine hesitancy?

Author Response

Response to reviewer 2:

The authors want to thank reviewer 2 very much for the useful comments and the rapid review process. We hope that we answered the questions in sufficient detail below.

Reviewer 2 wanted us to clarify if minors can choose to be vaccinated over parental objections and to discuss potential further implications.

It now reads (Line 301 revised manuscript): “It was surprising that only one third of participants knew about their right to decide to get vaccinated at age 14 [9] and that these minors could choose to be vaccinated over parental objections. This lack of awareness that schoolchildren do not need their parent’s consent to vaccination was neither associated with vaccine hesitancy nor with the intention to vaccinate against measles or HPV, but this may change in future if more schoolchildren are aware of their rights.”

Reviewer 3 Report

This is an interesting report on vaccination and disease knowledge and perceptions showing a fair level of vaccine hesitancy and an insufficient status of awareness about decision rights. I have several remarks:

L38 replace semicolon

L52 – probably the information could also be useful for regions facing a similar situation

L80 how were the youths themselves informed about the study?

L99 I wonder if the authors considered the fact that there may be a correlation structure between pupils from the same class taking part in the survey. Did you take this into account in the analysis, or a sensitivity analysis? As it seems there is no information about the specific class, this would hinder such an approach.

L 100 how was the approach to (partially) missing data? Were all surveys filled out completely? In line 314 you mention this, but this is rather late and not expanded on any further.

L 109 thank you for very clear definitions, however under unknown vaccination I wonder if it should be “did not recall”

L 173 I do not think these are relative risks in an epidemiologic terminology sense. Please review, also for the further use of RR in the manuscript (under determinants). A clarification would do, since use of the generic term relative risk usually needs some specification. The use of both PR and RR is not well differentiated and should also be explained better, if maintained.

L 170 intention to vaccinate against influenza- it is hard to interpret the data around this issue since readers may not be aware of the vaccination policy towards influenza in Austria. If it is not recommended for this age group, and analysis of intention to vaccinate may be misleading. Perhaps an overall table of vaccination policies for the age group in Austria may be included for clarity.

L 257 the question on trusted sources seems to have been posed such that a cumulative 100% was possible. This leads to surprisingly low figures for everything beyond doctors. How does this compare to other studies on preferred information sources.

Discussion:

My impression from reading the survey that there is quite a good level of information on many, not all aspects of vaccination and related diseases. For example, specific information on measles vaccination type was surprisingly high. So what do the authors take from their survey, does it corroborate the low health literacy seen in the respective HL survey? The conclusion seems to highlight low knowledge levels, but without comparisons to other regions or countries one can argue differently as well.

The limitations should also include the issue of correlated data as indicated above.

Are there any approaches beyond school lectures to be recommended to reduce vaccine hesitancy?

English language – needs some revision

Author Response

Response to reviewer 3:

The authors want to thank reviewer 3 very much for the useful comments and the rapid review process. We hope that we answered the questions in sufficient detail.

Old line L38: The semicolon was replaced by comma

Old line L52: the comment was included and it now reads (line 54 of the revised manuscript): “The information obtained from this survey will be a useful tool in tailoring immunization strategies in the region and could also be useful for other regions facing a similar situation”.

Old line L80: The schoolchildren were informed by their teachers about the study. It now reads (line 83 of the revised manuscript): “The potential participants were informed by the responsible teachers about the study. They were provided with an information letter including an informed consent, which parents were encouraged to sign.  Schoolchildren then brought the signed informed consent back to school.”

Old line L99: As mentioned by the reviewer, we did not obtain the information about the class in our dataset due to data confidentiality issues and therefore it was not possible to conduct a sensitivity analysis. We included the issue of correlation structure in the discussion as follows: (line 326, revised manuscript): “Another important limitation of our study was that many participants did not respond to all questions. This limited the power of the study and thus could have masked other significant associations. Furthermore, we could not conduct a sensitivity analysis to investigate potential correlation structures within each class as data on classes were not obtained.”

Old line L100: the approach to partially missing data was included in the methods section (line 101 of the new manuscript):  “Partially missing data were excluded from the analysis”. We also further expanded on this issue in the discussion section (line 326, new manuscript): “Another important limitation of our study was that many participants did not respond to all questions which limited the power of the study and thus could have masked other significant associations”.

Old line L109: Reviewer 3 suggested to change the category “unknown vaccination status” to did not recall: As this category may also contain persons who did not want to reply to the question we would prefer leaving the classification. But the reviewer pointed out that there was a mistake in the manuscript, namely the “not” was missing. It now reads (line 119 of the new manuscript):” Unknown vaccination status: non-responders, persons who did not recall or did not want to answer the question about previous MMR or HPV vaccinations”

Old line L173: Thank you for this important comment. We fully agree and use the term prevalence ratio throughout the entire manuscript. The first change has been made in the methods section (Line 110 of the new manuscript) and it now reads: “We estimated the prevalence ratio the 95% confidence interval of potential determinants using the Taylor series in Epi.infoTM  between the vaccinated and unvaccinated group using univariate analysis.” We also changed RR to PR in table 3 (Lines 180, 217, 221 new manuscript), table 4 (Lines 223, 225, 227, 229, 230, 232 of the new manuscript) and table 5 (Lines 234, 238, 239, and 241 of the new manuscript) and table 6 (Line 256). The headings of table 3-6  now read “Association (prevalence ratio (PR) and 95% confidence interval (95% CI)) between of perception of severity of disease and intention to vaccinate against the disease”

Old line L170: Reply to the comment of vaccination recommendations in Austria. Instead of a table the following sentence was added in Line 38 of the new manuscript:  “ In Austria, all childhood vaccinations are recommended including influenza and most of them free of charge until age (except vaccination against meningococcal B, tick-borne encephalitis, varicella and influenza vaccination) “

Old line L257 It is true that there was only one single best answer for the most common trusted sources. In fact, there was a mistake in table 7. The data which should have been included in table 7 were written in the text below instead. We now changed the content of the table as it should have originally been and deleted the text which was below the table.

Table 7 should now read:

Table 7. Most common trusted sources of information (n=311)

Source of information

Proportion

95% CI

Medical doctor

80.7%

76.0-84.7

Family

7.1%

4.7-10.5

Public health authority including ministry of health

5.5%

3.4-8.6

Pharmacy

3.5%

2.0-6.2

Internet and social media

1.9%

0.9-4.1

Friends

1.0%

0.3-2.8

General media (newspaper and TV)

0.3%

0.1-1.8

 These results are consistent with published results among adults as it is stated in the discussion Lines 316-318:  “Only few participants reported having actively searched for information related to vaccination. Medical doctors were by far the most important and most trustworthy source, which is similar to published results for adults [27].”

Discussion:

The limitations now also include the issue of correlated data (Line 326-329: “Another important limitation of our study was that many participants did not respond to all questions which limited the power of the study and thus could have masked other significant associations. Furthermore, we could not conduct a sensitivity analysis to investigate potential correlation structures within each class as data on classes were not obtained”.

Other approaches beyond school lectures: Line 351, we added: “Furthermore, transparent, easy to understand and pro-vaccination messages should be distributed by social media.”

Regarding English language: we will take the publisher’s offer to support us with English language editing.

Round 2

Reviewer 3 Report

Adaquate response to my comments and questions.